# Abnormalities of EEG Functional Connectivity and Effective Connectivity in Children with Autism Spectrum Disorder

**DOI:** 10.3390/brainsci13010130

**Published:** 2023-01-12

**Authors:** Xinling Geng, Xiwang Fan, Yiwen Zhong, Manuel F. Casanova, Estate M. Sokhadze, Xiaoli Li, Jiannan Kang

**Affiliations:** 1School of Biomedical Engineering, Capital Medical University, Beijing 100069, China; 2Clinical Research Center for Mental Disorders, Shanghai Pudong New Area Mental Health Center, School of Medicine, Tongji University, Shanghai 200124, China; 3Department of Biomedical Sciences, University of South Carolina School of Medicine Greenville, 701 Grove Rd, Greenville, SC 29605, USA; 4State Key Laboratory of Cognitive Neuroscience and Learning, Beijing Normal University, Beijing 100859, China; 5College of Electronic & Information Engineering, Hebei University, Baoding 071000, China

**Keywords:** autism spectrum disorder, EEG, functional connectivity, effective connectivity

## Abstract

**Highlights:**

**What are the main findings?**

**What is the implication of the main finding?**

**Abstract:**

Autism spectrum disorder (ASD) is a heterogeneous neurodevelopmental disorder that interferes with normal brain development. Brain connectivity may serve as a biomarker for ASD in this respect. This study enrolled a total of 179 children aged 3−10 years (90 typically developed (TD) and 89 with ASD). We used a weighted phase lag index and a directed transfer function to investigate the functional and effective connectivity in children with ASD and TD. Our findings indicated that patients with ASD had local hyper-connectivity of brain regions in functional connectivity and simultaneous significant decrease in effective connectivity across hemispheres. These connectivity abnormalities may help to find biomarkers of ASD.

## 1. Introduction

Autism is a heterogeneous neurodevelopmental disorder that affects normal brain development [1]. Brain connectivity is one of the ways to investigate potential biomarkers for autism [2]. A recent upsurge in interest in the study of brain connectivity in autism indicates a paradigm shift away from understanding its biological role in affecting specific brain regions and toward imaging the brain’s overall connectivity pattern [3,4]. Additionally, growing evidence on the early development of white matter trajectories suggests overall connectivity as an early biomarker for ASD, as reported abnormalities appear in the first year of life [5,6].

Brain connectivity is a broad and multifaceted concept [7]. It can refer to the physical inter-connectivity (structural connections) of brain regions via axonal bundles, statistical dependence (functional connection) between time series of brain activity in various brain regions, or causal interaction (effective connection) between brain regions. Structural connectivity is typically assessed by deterministic or probabilistic fiber bundle imaging using diffusion-weighted images recorded by magnetic resonance imaging (MRI) scanners [8,9]. We can calculate the other two types of connectivity using EEG or magnetoencephalography [10]. A set of statistical dependencies between neural processes in the brain is referred to as functional connectivity. In its simplest form, the correlation of neuronal activities can be used to estimate functional connectivity, which is highly dependent on time (on the order of hundreds of milliseconds), although it cannot reveal causal links between distinct neural activities. By ascribing causality to dependency, effective connectivity attempts to explain or model the interaction between activated brain regions. Compared with functional connection, effective connectivity calculates directional connection [11,12].

Some previous studies have reported that the connectivity aspect of ASD was the combination of long-range insufficient connectivity and local excessive connectivity, EEG studies provided support for this conclusion with a decrease in EEG functional connectivity, which has been reported in many previous studies [13,14,15,16]. However, the results of local overconnectivity in ASD were not so consistent. A few studies reported local overconnectivity [17,18,19], while others reported local underconnectivity in ASD [20]. Some studies have also reported the abnormal functional connectivity of ASD at different frequency bands. Domínguez reported that in the alpha, theta, and delta frequency bands, children with ASD exhibited increased connectivity when using EEG coherence compared with TD children [21]. However, Boersma et al. reported that the EEG connectivity of children aged 2−5 years was not significantly different between the ASD and control groups in 1−30 Hz using a phase lag index [22]. Other studies have discovered no significant changes in functional EEG hemisphere connectivity in the gamma band between the low-risk and high-risk infants [23], and there was no difference in connectivity between 6-month-old infants [24].

In summary, while long-range underconnectivity of ASD was reported, it was likely that this was related to the task requirements. The reasons for the lack of consistency in functional connectivity results might be owing to the use of different variables in studies, such as age, the frequency band of interest, selected functional connectivity index, and different sample sizes. Additionally, the clinical heterogeneity of ASD and its subgroups may be the source of the contradictory outcomes.

Effective connectivity, as opposed to functional connectivity, calculates directionality. The majority of previous studies that used effective connectivity methods established brain networks based on the analysis of relevant algorithms [25]. Some large sample studies were primarily based on the functional MRI database [26,27]. Few studies have analyzed the differences in effective connectivity between children with autism and healthy children using EEG [28]. We selected this study owing to the discrepancy between the research mentioned results on brain functional connectivity between children with ASD and healthy children and the research gap regarding differences in effective connectivity. Resting-state activity is a major factor determining other, more particular, responses to stimuli [29], we used the resting-state EEG data samples of 179 children with autism and healthy children to explore the differences in brain connectivity between them, providing a scientific basis and objective indicators for investigating the biomarkers of brain connectivity of autism.

## 2. Materials and Methods

### 2.1. Participants

A total of 179 children aged 3−10 years were enrolled in this study. They were divided into a typically developing (TD) children group and an ASD group. There were no statistical differences in age or gender between the two groups, and the participation information is shown in Table 1. All the TD children were recruited from a local kindergarten, and the inclusion criteria were as follows: (1) the child guardian agreed to and signed the informed consent form; (2) the children did not have any mental disorders or history of autism and/or other mental diseases; (3) the children did not have any nervous system diseases or other serious physical diseases, and there was no history of severe brain trauma or febrile seizures; (4) all the children were right-handed. All participants with ASD were diagnosed by experienced Chinese psychiatrists using the psychoeducational profile (Third Edition) and Diagnostic and Statistical Manual of Mental Disorders–V criteria [30]. The inclusion criteria were as follows: (1) informed consent was obtained from the children’s parents prior to participation; (2) children with ADHD, RETT syndrome, growth retardation, and other brain developmental disorders were excluded; (3) there were no serious physical diseases, history of severe brain trauma, or history of febrile seizures, and all children were right-handed; (4) except for drugs administered during ordinary behavior training, children did not receive any psychiatric medicines. This study was conducted per the Declaration of Helsinki and approved by the ethics committee of Beijing Normal University.

### 2.2. Data Acquisition


EEGs were recorded in a shielded room where children sat on comfortable chairs with their eyes open. All children washed their hair and scalp before EEG data acquisition. We welcomed the children with ASD to the laboratory ahead of time to familiarize them with the surroundings. We made every effort to keep everyone quiet and to keep movements to a minimum. However, autistic children were more difficult to stay still for a while, and EEG data will be seriously interfered, so we have made careful data preprocessing. The EEG was recorded for approximately 5−10 min. Data were recorded using a 128-channel HydroCel Sensor Net System (Electrical Geodesics, Inc., Eugene, OR, USA). The electrode impedance was kept below 50 KΩ throughout EEG acquisition. The sampling frequency was 1000 Hz, and the reference electrode was Cz.

### 2.3. Data Preprocessing

MATLAB R2016a and EEGlab V13.5.4b were used for offline data analysis.

After downsampling EEG data to 200 Hz, a 1−45 Hz bandpass filter was employed to preprocess the EEG signal, which was then segmented to 10 s epochs. To remove eye blink, muscular artifact, and electromyogram, independent component analysis (ICA) algorithm was employed. Finally, all channels were re-referenced to an average reference, and 62 electrodes (black dots in Figure 1) were selected, mainly according to the 10-10 electrode system, for subsequent analysis [31]. At most, 10 epochs were used to for each subject in the following analysis.

### 2.4. Functional Connectivity

For the preprocessed EEG signals, the brain functional connectivity was assessed to determine the synchronization degree of EEG time series in corresponding brain regions, using the weighted phase lag index (wPLI). Stam et al. proposed the calculation method of PLI to improve phase synchronization [32] based on the notion that the non-zero phase synchronization between two signals cannot be interpreted by the same source signal through volume conductivity, thus reflecting the real synchronization between two systems. However, the sensitivity of PLI to noise and volume conduction may be affected by its exponential discontinuity since small disturbances will change the phase lag into lead and vice versa. Small amplitude synchronization effects make this more challenging. Therefore, to better detect real changes in phase synchronization and reduce the impact of non-correlated noise, weight was introduced into the phase synchronization index based on the imaginary part of a cross-spectrum used to obtain wPLI [33].

We defined a time series s(t), and the corresponding analytic signal was as follows:(1)$$z\left(t\right)=s\left(t\right)+jℋ\left[s\right]\left(t\right)=A\left(t\right)e^{j\phi \left(t\right)}$$
where A(t) was the instantaneous amplitude of signal s(t) and ϕ(t) was the instantaneous phase of s(t).
(2)$$\phi \left(t\right)=\arg\left[z\left(t\right)\right]=\arctan\left(\frac{ℋ\left[s\right]\left(t\right)}{s\left(t\right)}\right)$$
where the definition of Hilbert Transform of signal s(t) was:(3)$$ℋ\left[s\right]\left(t\right)=\frac{1}{\pi }P.V.\int_{-\infty }^{\infty }\frac{s\left(\tau \right)}{t-\tau }d\tau$$
where *P.V*. represents the integral in the sense of Cauchy principal value. $\phi_{1}\left(t\right)$ and $\phi_{2}\left(t\right)$, respectively, represent the instantaneous phase of the two signals; thus, the phase difference of the two signals △ϕ(t) was as follows:(4)$$△\phi \left(t\right)=\phi_{1}\left(t\right)-\phi_{2}\left(t\right)$$

PLI was calculated as follows:(5)PLI=|⟨sign[△ϕ(t)]⟩|

wPLI measures the distribution of the phase angle difference between two time series to the positive or negative part of the imaginary axis in the complex plane, which is defined as follows:(6)$$\mathrm{wPLI}=\frac{\left|E\left\{\left|\xi \left\{X\right\}sgn\left(\xi \left\{X\right\}\right)\right|\right\}\right|}{E\left(\left|\xi \left\{X\right\}\right|\right)}$$
where *X* is the cross-spectrum of two time series, ξ{X} represents the imaginary component of the cross-spectrum, and the calculation based on the imaginary component only increases the robustness to noise. The value of wPLI was 0−1. 

Based on the maximum space coverage, we selected five brain regions to calculate their connectivity, including the frontal lobe (F), left temporal lobe (LT), parietal lobe (P), right temporal lobe (RT), and occipital lobe (O). Eight brain regions were selected to calculate transhemispheric connectivity, including the left frontal lobe (LF), right frontal lobe (RF), left temporal lobe (LT), left parietal lobe (LP), right parietal lobe (RP), right temporal lobe (RT), left occipital lobe (LO), and right occipital lobe (RO). Figure 1 presents a schematic diagram of the functional connectivity in brain regions and transhemispheric connectivity. 

In this study, we calculated the cross-spectrum of two preprocessed EEG epochs, and then calculated the wPLI value in four canonical frequency bands, that is delta (1–4 Hz), theta (4–8 Hz), alpha (8–13 Hz) and beta (13–30 Hz). Then, with regard to the functional connectivity in one brain region, the wPLI was averaged among all electrode pairs in one brain region, and then averaged among all the epochs. On the other hand, the transhemispheric connectivity was averaged among the wPLI values of all the electrode pair in two different brain regions and then averaged among epochs.

### 2.5. Effective Connectivity

The causal link between EEG signals, i.e., effective connectivity, was calculated using the direct transfer function (DTF). The DTF algorithm was based on the following Granger causality hypothesis proposed by economist Granger [34]: Two time series X and Y were supposed. The accuracy of using the time information before X to predict that after X was recorded as M, and the accuracy of using the time information before X and Y to predict that after X is recorded as N. If M is greater than N, it indicates that time series Y will impact time series X, and X Y are believed to have Granger causality. This method can also be used to explore the dynamic causal relationship between different time series.

The following multivariate autoregression model for 62-channel EEG signals was established:(7)$$X_{t}=\sum_{i=1}^{p}A(i)X(t-i)+\varepsilon (t)$$
where p is the model order; A(i) is the 62×62 coefficient matrix; ε(t) is multivariate white noise. Equation (7) was Fourier transformed as follows:(8)$$X(f)=A^{-1}(f)\varepsilon (f)=H(f)E(f)$$
where f is frequency; and H(f) is the transfer matrix of the system, represented as follows:(9)$$H(f)=\;{\left[\sum_{i=1}^{p}A(i)e^{-2\pi fit}\right]}^{-1}$$

The connectivity strength between i and j was obtained by normalization, divided by all inflows of the channel i and the normalized DTF was regarded as a value between 0 and 1, representing the ratio of inflows from channel j to channel i. The definition formula was as follows:(10)$$DTF^{2}{}_{j\to i}\left(f\right)=\frac{|H_{ij}\left(f\right){|}^{2}}{\sum_{m=1}^{l}|H_{im}\left(f\right){|}^{2}}$$

Given that the directionality of effective connections is canceled out when averaged over brain regions, we only calculated effective connections across hemisphere brain regions. All channel EEG signals in the left and right frontal (LF and RF), left and right temporal lobes (LT and RT), left and right parietal lobes (LP and RP), and left and right occipital lobe (LO and RO) brain regions were averaged across the hemisphere. Since the MVAR model assumes that the signal is stationary, we used the data segmentation method to treat the EEG signal as quasi-stationary. Specifically, the preprocessed 62-channel EEG signal was divided into 10 s epochs (2000 data points). For each epoch, DTF was calculated in the frequency range of 1–30 Hz, and the frequency step was 1 Hz. 

A surrogate data set was constructed by randomly disrupting the phase on each channel of this multichannel data. Surrogate times were set to 100. Then, DTF_surrogate_ was calculated on this multi-channel surrogate data, and the obtained 100 DTF_surrogate_ values were arranged from largest to smallest. Using the fifth DTF_surrogate_ value as the threshold (significance level is set to 0.05), the DTF value calculated from the EEG signal (DTF_EEG_) was compared with this threshold. If DTF_EEG_ was greater than this threshold, it is considered to have a significant causal relationship. Finally, all significant DTF_EEG_ values were averaged across all epochs and four canonical frequency bands and used as the effective connectivity in transhemispheric regions.

### 2.6. Statistical Analysis

Statistical analyses were performed using MATLAB software (Mathworks Corp., Natick, MA, USA). A paired two-sample *t*-test with false discovery rate (FDR) adjustment for multiple comparisons was conducted to identify the differences of wPLI and DTF values between the TD and ASD groups. *p* < 0.05 was considered statistically significant.

## 3. Results

### 3.1. Functional Connectivity Differences in Two Groups

In the four frequency bands, the connectivity in most of the brain regions of the ASD group was higher than that of the TD group. The frontal lobe (*p* = 0.0257, *t* = 208344) and parietal lobe (*p* = 0.0389, *t* = 2.6923) of the theta band revealed significant differences, and the left (*p* = 0.0055, *t* = 3.1912) and right temporal lobes (*p* = 0.0206, *t* = 2.9077) of the beta band exhibited significant differences, as presented in Figure 2. Simultaneously, we calculated the transhemispheric connectivity of the two groups. No significant differences were observed in the statistical analysis. In some bands, the connectivity of some brain regions in the ASD group was higher than that in the TD group, such as the left frontal lobe and the right temporal lobe in the delta band, the left parietal lobe and the right temporal lobe in the theta band, the left frontal lobe and the right parietal lobe in the alpha band, and the left parietal lobe and the right occipital lobe in the beta band. However, in some frequency bands, connectivity in the ASD group was lower than in the TD group. For instance, the left temporal lobe and the right parietal lobe in the delta frequency band, the left parietal lobe and right occipital lobe in the theta frequency band, the left occipital lobe and right temporal lobe in the alpha frequency band, and the left frontal lobe and right parietal lobe in the beta frequency band. The results are presented in Figure 3, considering the transhemispheric brain region connectivity in the alpha frequency band as an example.

### 3.2. Effective Connectivity Differences between the Two Groups

We calculated the DTF values between the ASD and TD groups in the transhemispheric brain region. These results after FDR multiple test correction are depicted in Figure 4. Based on the bar chart, the blue block indicates that the DTF value of the ASD group is lower than that of the TD group in the delta, theta, alpha, and beta frequency bands. The results reveal that the effective brain connectivity of the ASD group in the transhemispheric brain region was lower than that of the TD group.

## 4. Discussion

Although the autistic brain has been reported to exhibit a pattern of long-range underconnectivity and local overconnectivity, the results remained unreproducible and conclusions were divergent regarding the nature of altered connectivity in ASD. In this study, we focused on resting-state EEG, because the fundamental importance of the ongoing nervous system activity has been widely recognized today, and a more in-depth investigation of brain activity in periods of minimal sensory perturbation has been advised, as it may provide the best opportunity to study the intrinsic connectivity of the brain [29]. We found robust differences between ASD and TD children during a resting state, reflecting contrasting patterns of over- and underconnectivity.

### 4.1. Analysis of Functional Connectivity Differences between the ASD and TD Groups

Our findings revealed an increase in functional connectivity of the frontal and parietal lobes of the delta band in brain regions of children with autism, which is consistent with previous results. The lack of dorsolateral prefrontal lobe inhibition may be linked to the increase in frontal connectivity. The increase in parietal lobe connectivity may be related to the attention deficit and hyperactivity disorder linked to children with autism. This finding suggested that the autistic brain may be more prone to processing local information owing to the imbalance of excitation and inhibition of local neurons in the brain. Our findings also confirmed Casanova’s previous hypothesis that the autistic brain has relatively smaller, but a greater number of, microcolumns, which propitiates an excess of shorter connections (e.g., arcuate fibers) [35]. Simultaneously, our results prove that the common cognitive impairment of autism may be attributed to their increased neural synchronization in the slow wave band.

Our study also demonstrated that the functional connectivity of the left and right temporal lobes of the brain in the ASD group was higher than that of the TD group in the beta frequency band. High-frequency priority is associated with more localized processes, whereas activity priority in the lower-frequency band is associated with broader comprehensive processes [36,37]. Top-down comprehensive processes involving long-range connectivity (the process of creating perception by fusing previous knowledge of the world with the incoming signals from the senses) are usually associated with slower rhythms (delta and theta) [38], whereas local synchronization is typically related to faster frequencies (beta and gamma) through the cortical network of the bottom-up process (the process of modifying the internal representation of the world to reduce its discrepancy with sensory data) [39]. However, since functional connectivity in the high-frequency band has not previously been studied, our findings served as a basis for this judgment. Our research findings did not exhibit significant differences in functional connectivity between the ASD and TD groups across hemispheric brain regions. A possible explanation for this is that our subjects were children, and their brain development was not complete. Most previous studies on functional connectivity across brain regions were conducted on adults with autism. Their conclusions were inconsistent owing to the variations in the quantity and aptitude of subjects. Our findings supported the notion that hemispheric brain regions in children with autism have aberrant functional connectivity.

### 4.2. Analysis of Effective Connectivity Differences between the ASD and TD Groups

Currently, the analysis of effective connectivity is mostly focused on fMRI research, and limited research has been performed using EEG to calculate autism effective connectivity. In this study, a DTF algorithm was used to calculate effective connectivity. The results revealed that the effective connectivity across brain regions of children in the TD group was higher than in the ASD group in all four frequency bands (delta, theta, alpha, beta), which is consistent with findings provided by fMRI in previous studies [40]. The aberrant direction of information flow between brain regions of children with autism and the lower effective connectivity impeding information interchange in their brains were revealed by our results.

## 5. Limitations of This Study

Although this study suggested differences between the ASD and TD groups, there were still some limitations. (1) This connectivity analysis was conducted in electrode-based brain regions, and source localization tool was not used. (2) There was a lack of EEG data on infants and adolescents, meaning it was not possible to complete the difference analysis for all age groups. (3) There was a lack of comparison between different connectivity calculation methods. (4) Sophisticated statistical models should be used to analyze the contribution of age to the differences in connectivity and development patterns. (5) Some autistic children did not complete the usual confirmatory instruments, such as the ADOS. Our subjects might include patients with many genetic disorders, which could have affected the results. Further work is needed to better understand the complex interactions between frequency bands and brain regions, and how they relate to different cognitive processes. Suitable event-related protocol will need to be devised in a principled way and for the recruitment of subjects, we would confirm their clinical information more carefully.

## Figures and Tables

**Figure 1 brainsci-13-00130-f001:**
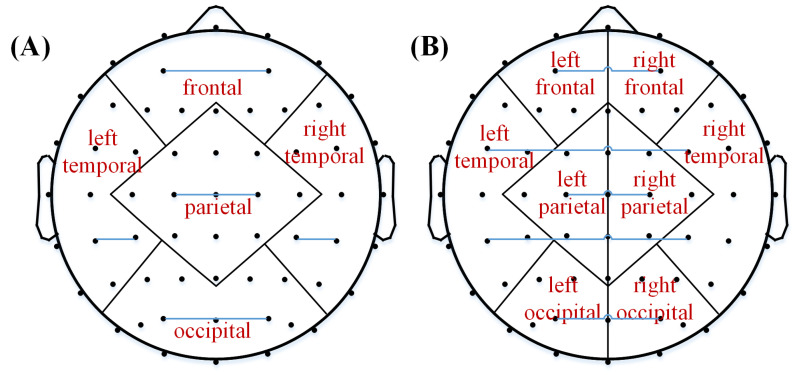
Schematic diagram of the functional connectivity in brain regions (**A**) and transhemispheric connectivity (**B**). The blue line denotes the connectivity between the electrode pairs as examples.

**Figure 2 brainsci-13-00130-f002:**
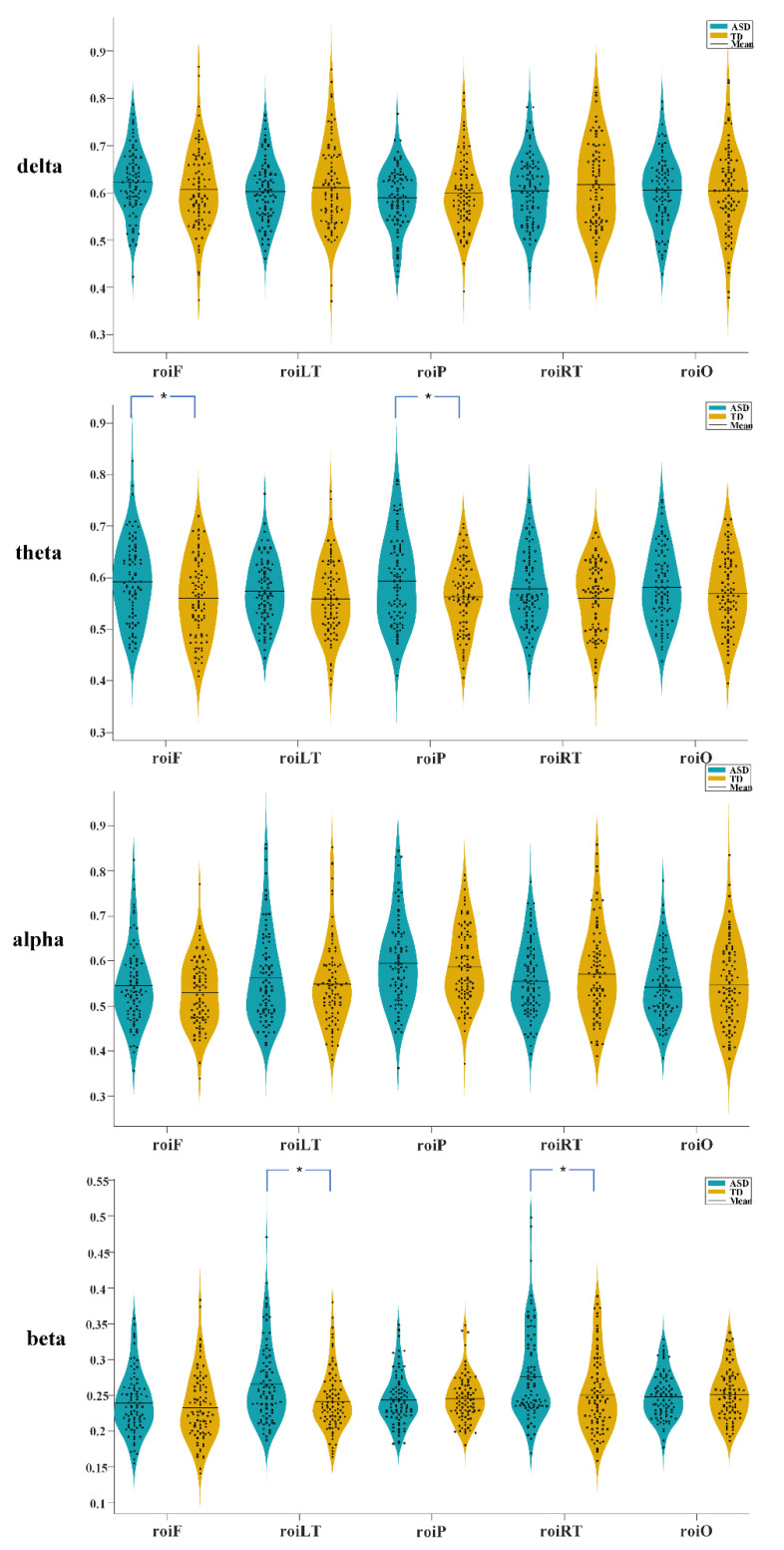
The functional connectivity differences in brain regions of four frequency bands between the au-tism spectrum disorder (ASD) and typically developing (TD) groups (* *p* < 0.05).

**Figure 3 brainsci-13-00130-f003:**
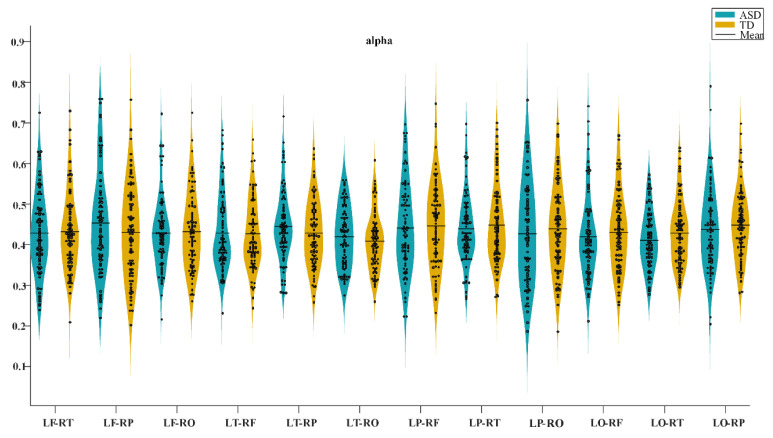
The functional connectivity differences in transhemispheric brain regions of the alpha band between autism spectrum disorder (ASD) and typically developing (TD) groups.

**Figure 4 brainsci-13-00130-f004:**
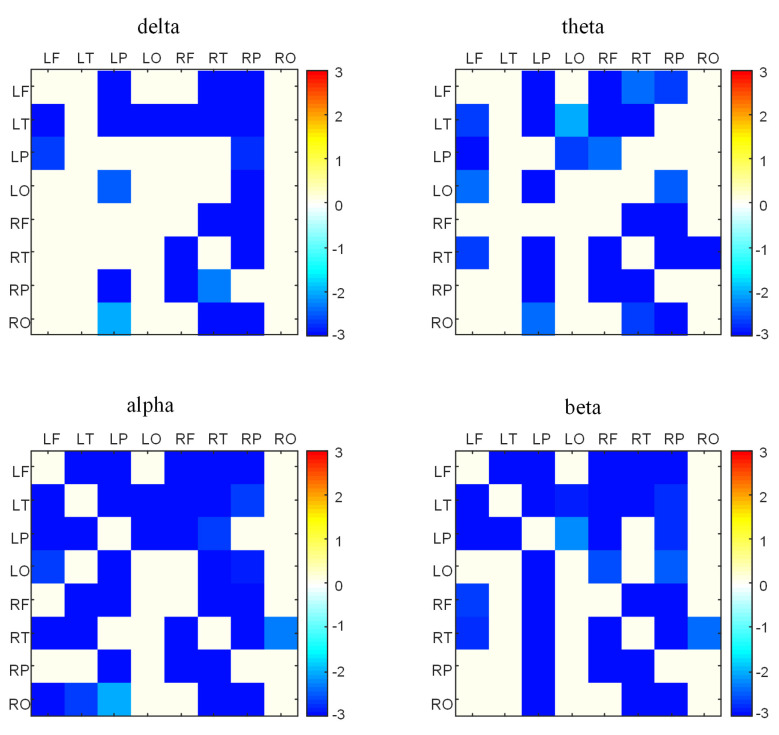
The statistical comparison of effective connectivity in transhemispheric brain regions between autism spectrum disorder (ASD) and typically developing (TD) groups. The color bar denotes the *t* value after FDR correction.

**Table 1 brainsci-13-00130-t001:** Participant information.

	TD (*n* = 90)	ASD (*n* = 89)	Differences between Two Groups
Age	5.09 ± 2.27	5.17 ± 1.89	*t* = −0.497, *p* = 0.638
Male/Female	76/14	71/18	$\chi^{2}=1.207$, *p* = 0.359

## Data Availability

The datasets generated and analyzed in this study are available from the corresponding author upon reasonable request.

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
