# Peer review of "Abnormalities of EEG Functional Connectivity and Effective Connectivity in Children with Autism Spectrum Disorder"

_brainsci, 2023, doi:10.3390/brainsci13010130_

Round 1
Reviewer 1 Report
In their paper, Geng et al. assessed the functional and effective connectivity in patients with autism disorder. The functional connectivity was measured using weighted phase lag index and effective connectivity was assessed with directed transfer function. The results showed patients with autism had higher local connectivity and decreased effective connectivity across different hemispheres. After thoroughly reading the manuscript, I have some minor concerns. I would propose the minor revision of this manuscript and please see my specific comments below.
Minor:
Why did the authors selected 62 electrodes from 128-channels recording? Please cite any reference for the selected electrodes. What does blue lines represent in Figure 1A? And why frontal and occipital lobes were not divided into left and right hemispheres?
Please provide the descriptive stats for the length of EEG recording? And how many epochs were included in the analysis?
Please include the t-values for functional connectivity results.
The frequency bands were overlapping. Is this how analysis was done?
Line 193: ….Equation 8 was transformed….
Reviewer 2 Report
1. Line 12
Please indicate which author is affiliated to the sixth affiliation.
2. Line 44–45
This description might be inappropriate, as there are also numerous studies that used fMRI data to conduct functional or effective connectivity analyses (e.g., Friston, K. J. (2011). Functional and effective connectivity: a review. Brain Connectivity, 1(1), 13-36.).
3. Line: 54
Is it long-term or long-range?
4. Line 58
Please provide the references corresponding to the statement “…while others reported local under-connectivity or both patterns in ASD”.
5. Line 76–79
Please provide the references corresponding to the statement “Some large sample studies were primarily based on the functional MRI database. Few studies have analyzed the differences in effective connectivity between children with autism and healthy children using EEG.”
6. Line 87–107 & Table 1
ASD are a group of complex and heterogeneous disorders involving multiple neural system dysfunctions. Like other neurodevelopmental disorders, people with ASD may have different symptom distribution as well as levels of severity. And these have been reported to bring out significant impacts on the connectivity measurements (Please see your reference #14, #17, & #18 for examples). Therefore, in order to have a clear profile of the ASD group adopted in the current study, please provide the information regarding clinical diagnosis (i.e., high/low function autism, Asperger, & PDD-NOS), symptom severity (e.g., scores from Autism Diagnostic Observation Schedule or Autism Diagnostic Interview), and IQ in patients’ demographics. Also, I recommend the authors to examine the relation between these demographics and the various connectivity measurements they have, as it could help to further elucidate the pathological mechanisms of ASD.
7. Line 282
Please provide the corresponding references to Casanova’s previous hypothesis.
8. Line 289–297
Please provide the corresponding references to these statements “High-frequency priority is associated with a more localized process whereas activity priority in the lower-frequency band is associated with a broader comprehensive process. The top-down comprehensive process involving long-range connectivity (the process of creating perception by fusing previous knowledge of the world with the incoming signals from the senses) is usually associated with slower rhythms (delta and theta), whereas local synchronization is typically related to faster frequencies (beta and gamma) through the cortical network of the bottom-up process (the process of modifying the internal representation of the world to reduce its discrepancy with sensory data).”
Reviewer 3 Report
In the manuscript, entitled" Abnormalities of EEG functional connectivity and effective connectivity in children with ASD." by Dr. Kang et al. highlighted EEG biomarkers of ASD. Although the scientific premise for the paper is exciting and novel, but without source localization and the difference analysis of all age group put??? On the study outcome. The study is well conducted, and draft is well written.
I have some major concerns
The paper did not explain in detail how the computational method is constructed and applied. The finding is not reliable as the sample size is small.
ASD is a heterogeneous group of disorders. By selecting autism based only on DSM you no doubt have a very heterogeneous group of subjects Probably many would not even qualify for the diagnosis based on the usual confirmatory instruments such as the ADOS. While you list a few exclusions (i.e. developmental delay), may be you have included those with many genetic disorders which are not tested for…
Round 2
Reviewer 2 Report
The authors have nicely responded to almost all the questions I raised before. Only one remaining issue needs to be addressed, which is that the ref. #36 is actually an HRV (heart rate variability) study and is not likely to support the authors' statements regarding the nature of EEG oscillatory activities (line 290-292).
